# The Use of Fermented Plant Biomass in Pigs Feeding

Barbara Płacheta [1],*, Ilona Motyl [2], Joanna Berłowska [2] and Marta Mroczyńska-Florczak [2]

1   Department of Environmental Biotechnology, Faculty of Biotechnology and Food
    Science, Interdisciplinary Doctoral School, Lodz University of Technology, 171/173 Wólczańska Street,
    90-924 Lodz, Poland
2   Department of Environmental Biotechnology, Faculty of Biotechnology and Food Science,
    Lodz University of Technology, 171/173 Wólczańska Street, 90-924 Lodz, Poland
*   Correspondence: barbara.placheta@dokt.p.lodz.pl

**Abstract:** The demand for animal-based food production is increasing mainly due to the rapid growth of the human population. The effective production of high-quality agricultural products promotes and protects the natural environment, human health, and animal welfare. Sustainable processing involves minimizing the waste stream. One way to use agricultural plant-based waste, which is often rich in bioactive substances, is to produce fermented feed in accordance with the principles of sustainable development. Corn, yellow lupins, and narrow-leaved lupins are rich in nutrients, and are suitable for fermentation and use in pig feed. They are also safe for weaned piglets. Used as a feed additive, fermented plant biomass has a positive effect on the health of pigs, increasing their weight and improving the taste and appearance of the meat. The fermentation of plant biomass reduces antinutritional substances that are abundant in feed components. It also improves the digestibility of the silage and the composition of the pig's intestinal microflora.

**Keywords:** fermentation; plant waste biomass; maize; lupine; pigs feeding; sustainable agriculture

## 1. Introduction

### 1.1. Fermented Biomass as an Element of the Circular Economy

Agriculture is the most vulnerable sector of the economy to climate change. Changes in precipitation, temperature, $CO_2$ concentrations, and sea levels, as well as the increasing intensification and frequency of extreme weather events (including heat stress and its effects), significantly affect the quantity and quality of crops [1]. Sustainable agriculture involves the efficient production of high-quality, agricultural products. The purpose is to protect and promote the natural environment, ensuring animal health and welfare, as well as sustainable social and economic conditions for farmers, workers, and local communities. In line with the principle of sustainable development, as defined by the 1987 World Commission on Environment and Development, economic systems should be adjusted as soon as possible to counteract the effects of climate change [2]. The Common Agricultural Policy (CAP) for European Union member countries defines nine key social, environmental, and economic goals, including combating climate change, caring for the environment, and protecting health and food quality [1].

Population growth requires the production of more and more food, which results in more agro-industrial waste and economic losses. Often, production residues are rich in active compounds that can be used as additives in food and functional food, animal feed, pharmaceuticals, cosmetics, or bio-packaging. This not only reduces the amount of waste but also minimizes economic losses (Figure 1). Industrial fermentation processes can be used to produce biologically active compounds from agricultural waste. The spectrum of substrates and the metabolic abilities of microorganisms determine the final product [3]. The active compounds present in agricultural waste before fermentation include phenolic compounds, antioxidants, and compounds with anti-inflammatory and

anticancer properties. Vegetable and fruit waste is attracting increasing attention as a source of biomass, due to its quantity and high content of unused nutrients [4,5].

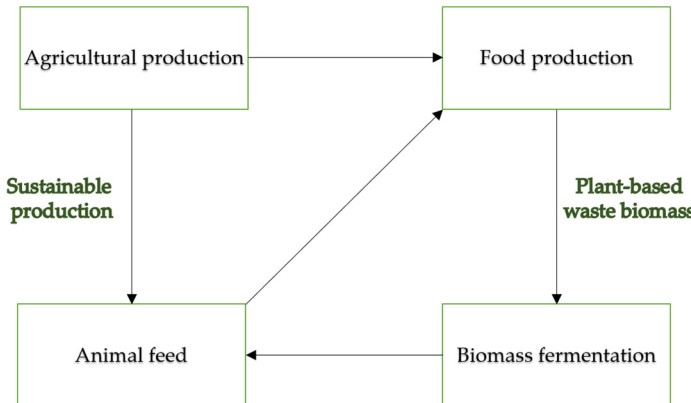

**Figure 1.** Fermented biomass as an element of the circular economy.

### 1.2. Contemporary Models Related to Pig Feeding

According to UN predictions, the human population is expected to reach almost 10 billion by 2050. Population growth is creating increased demand for meat products. At the same time, consumer standards are rising. Meat obtained by breeding animals, including pigs, must meet higher expectations, not only in terms of quality but also of appearance. This demand creates a shortage of animal feed, and contributes to the growing interest in the modification of plant waste into wholesome animal food [2].

Increased meat production due to the growing population also has serious consequences for the climate. There are two ways in which animals can be fed and produce meat sustainably. The first method of meat production is based on the requirements of the animal. It involves the precise feeding of animals with genetically modified feed, produced with the use of improved production methods. In this method the animals are monitored for disease and welfare. However, this system may lead to lower animal welfare standards and contribute to a decline in animal immunity. This method is characterized by a high input and maximum efficiency, based on the uniform production of the protein on the smallest possible surface in such a way as to have a minimal impact on the environment. In addition, the potential loss of nutrients, the release of manure into the wastewater, and the production of large amounts of greenhouse gases can contribute to the opposite effect—meat production is unsustainable [2,6].

The second system emphasizes environmental conditions and the availability of raw materials for feed, allowing local feeds, feedstuff co-products, or food waste to be fed to animals. This system assumes a reduced input and reduced production, as it is based on the selection of animals that are more resistant to climate change and are able to convert low-quality feed into meat [6]. Experts believe that the use of the second system will be less economical due to reduced yield forecasts, the potential costs of reorganizing the feed supply chains, the need to supplement unsustainable nutrient quality, or the pretreatment of feed to, for example, reduce antinutritional factors, which will directly increase meat prices for consumers. Therefore, the best solution is to combine the two systems by assessing the amount of available feed and by-products in the local area, identifying available areas for plant and livestock production, setting production levels, and controlling production via established national and international bodies [2,6].

According to Regulation (EC) No 1829/2003, genetically modified organisms cannot be used for food and/or feed if they are not covered by the EU authorization or if the conditions specified in this authorization are not met. The European Union member states (including Poland) are planning to introduce an order to limit the use of genetically modified ingredients in animal feed in the near future. This will exclude the use of feed with

the addition of soy. High-protein alternatives are therefore being sought. One possibility is to use legumes grown in Poland and Europe [7,8].

The diet of pigs, regardless of their age, is based on high-protein ingredients with a good content of amino acids (including lysine, methionine, and cysteine), such as yellow lupine or genetically modified soybeans. Pig feeding is a complex process, requiring specific methods for each stage of the pig's development. The most sensitive stage is weaning piglets from sows and subsequent rearing. To minimize quantitative and economic losses, the breeder should take care to implement appropriate conditions and processes, including enzyme training, feeding, water availability, temperature, appropriate early feeding (when feeding the sows and taking care that the piglets do not eat the sow's feed), feed acidification, the removal of uneaten feed, and veterinary control. Pigs reach full development of the digestive tract at the age of 5–6 months. Therefore, from the 5–7th day of life, piglets are given prestarters to accelerate their development, stimulating digestive processes and helping them learn how to feed independently [6,9,10].

## 2. Fermented Biomass as a Feed Component

There is increasing interest in using fermented feed as food for pigs as an alternative to antimicrobial growth promoters. Fewer and fewer microbes react to the action of antibiotics, and resistance to antibiotics may also pose a threat to consumers. Probiotics are defined by the WHO and FAO as 'living organisms which, when administered in appro-priate amounts, benefit the health of the host.' In the case of pigs, they can promote growth and nutrient utilization, modulating their gut microflora. The use of probiotic bacteria can also increase anti-infective properties against pathogens and decrease the amount of *Salmonella* sp. and *E. coli* in excreted feces. Feed fermented with probiotic microorganisms reduces the effects of stress in young pigs after weaning and weight gain, preventing weight loss and diarrhea, which may lead to economic losses. Probiotic microorganisms also provide greater fermentation efficiency [11,12]. Xu et al. [13] evaluated the effect of fermentation on the amount of feed nutrients required, pig growth efficiency, and meat quality. They showed that fermented feed significantly increases the crude protein content of the meat. Moreover, in weaned piglets and fattening pigs, fermented feed increases the digestibility of nutrients and improves daily weight gain. Fermented feed has a positive effect on the quality (nutrient content, including protein), taste, and appearance of meat, improving its marbling and reducing the content of water [13].

Fermentation is a process whereby water-soluble carbohydrates (WSC) are converted under anaerobic conditions into organic acids, mainly lactic acid, by lactic acid bacteria (LAB). This quickly lowers the pH of the ensiled biomass and interrupts the natural processes of nutrient decomposition by its own enzymes. Because the silage does not deteriorate, it can be stored for a long time, provided it has no contact with air. In an acidic environment, undesirable microorganisms that cause silage to rot and the protein to break down die, resulting in the formation of harmful substances. This slightly changes the chemical composition of silage compared with green forage: the concentration of sugars decreases and the content of neutral detergent fiber (NDF) increases, which may inhibit the absorption and use of other nutrients. Therefore, specific starters or fermentation modifications are applied. During forage ensiling, organic acids are used to ensure a sufficiently low pH and to protect against the proliferation of undesirable microorganisms, including *Enterobacteriaceae*. The population of microorganisms used for fermentation must produce sufficient amounts of organic acids to inhibit pathogens and pollutants [14,15].

There are diverse methods of plant biomass fermentation, which is influenced by the type of biomass, the starter used, and the process conditions Their course is influenced by the type of biomass, the starter used, and the process maintenance conditions (Table 1). The following factors influence the fermentation of plant materials: the dry matter content and chemical composition of the silage plant; the compatibility between the silage plant and the inoculated organism; the number and diversity of living microorganisms in the inoculum, as well as their ability to dominate the natural microflora of the plant; the method of

ensilage used; and the application of chemical and physical modifiers to the fermentation process and the aerobic stability of the silage [14,15].

To ensure the development of lactic acid bacteria, the raw material should have a high sugar content, proper moisture content, and be finely ground. The purity of the ensiled material should be ensured through compaction and by quicky shutting off access to air (covering). The most optimal conditions for the growth of lactic acid bacteria are 30–35% dry matter content, minimum 30% starch in dry matter, and maximum 20% crude fiber in dry matter [16]. When the biomass contains 70% water (30% dry weight), LAB are not restricted in their activity. A lower dry matter content of 30% can cause the growth of bacteria of the genus Clostridium. When the raw material is drier (more than 50% dry weight), LAB activity can be inhibited by 90%. Only 10% of LAB survive these extreme conditions, due to the lack of free water. During ensiling, the osmotic pressure in the ensiled biomass increases. The water available to bacteria in the dry raw material does not contain soluble carbohydrates, proteins, or minerals. Dry matter content above 50% makes it difficult to knead and remove air. This promotes the growth of aerobic bacteria and fungi. As much as 35% of dry matter means that some parts of the ensiled raw material contain up to 45% of dry matter [16–18].

**Table 1.** Selected methods of obtaining ensiled plant biomass.

| Type of Biomass | Microorganisms Used for Fermentation | Process Conditions | Results of Fermentation | References |
|---|---|---|---|---|
| Corn grains | *Lactobacillus fermentum*, *Saccharomyces cerevisiae*, *Bacillus subtilis* | Fermentation using a probiotic composition and NSP (nonstarch polysaccharides) with the activity of xylanase, β-glucanase, mannanase, cellulase, and pectanase. | Increased population of each microbial strain, increased fiber degradation, and increased protein contents; the residual contents of dry matter, crude ash, and reducing sugar decreased. | [19] |
| Corn bran | *Bacillus subtilis* MA139, *Saccharomyces cerevisae* | Fermentation for 14 days at 30 °C. | Decreased amount of cellulose, hemicellulose, and lignin; increased amount of soluble dietary fiber and nonstarch polysaccharides as well as arabinose, xylose, and glucose. | [20] |
| Narrow-leaved lupine | *Candida utilis* | Fermentations were carried out under aerobic conditions (natural pH = 5.5) for 24 h in a continuous mixing system. Then, yeast enzymes were deactivated for 10 min at 70 °C. | Increased contents of alkloids, protein, lysine, cystine, and threonine contents but not the methionine. Fermentation also reduced the Acid Detergent Fibre (ADF), NDF, and phytate-P. Fermentation significantly improved the digestibility of protein, asparagine, threonine, tyrosine, histidine, and arginine. Fermented products were characterized also by acidic pH and higher content of yeast and bacteria. | [21] |
| | *Saccharomyces cerevisiae*, *Saccharomyces carlsbergensis* | Aerobic conditions (natural pH = 5.5) for 24 h in a continuous mixing system. Then, yeast enzymes were deactivated for 10 min at 70 °C. | The content of crude ash and ADF significantly increased in all fermented products, whereas the ether extract and Nitrogen-Free Extract (NFE) contents significantly decreased. | [22] |

**Table 1.** *Cont.*

| Type of Biomass | Microorganisms Used for Fermentation | Process Conditions | Results of Fermentation | References |
|---|---|---|---|---|
| Yellow lupine | *Saccharomyces cerevisiae*, *Saccharomyces carlsbergensis* | Aerobic conditions (natural pH = 5.5) for 24 h in a continuous mixing system. Then, yeast enzymes were deactivated for 10 min at 70 °C. | The content of crude ash and ADF significantly increased in all fermented products, whereas the ether extract and Nitrogen-Free Extract (NFE) contents significantly decreased. The metabolizable energy was similar in all the samples. | [22] |
| Soy | *Rhizopus microspores zm. microsporus* LU 573 | Soybeans were soaked overnight in tap water, then washed with tap water and boiled for 20 min in fresh tap water at a 1:3 ratio. The cooked seeds were cooled and dried at room temperature, inoculated with a 7-day suspension containing *Rhizopus microsporus zm. microsporus* LU 573, 0.85% HCl, and 0.1% peptone, then fermented for 72 h at 30 °C. | Increased contests of crude lipid and crude protein in cooked and fermented soya beans. No differences in crude lipid and crude protein between cooked and fermented soya beans, but values for crude fibre were higher for fermented soya beans. Fermentation led to a major increase in nonprotein nitrogen. | [23,24] |
| | *Rhizopus microspores zm. microsporus* LU 573 *Bacillus subtilis* LU B83 | Fermentation was carried out for 48 h at 37 °C in large vessels containing ± 35 kg of inoculated cooked soybeans. | Increased contests of crude lipid and crude protein in cooked and fermented soya beans. No differences in crude lipid and crude protein between cooked and fermented soya beans, but values for crude fibre were higher for fermented soya beans. Fermentation led to a major increase in nonprotein nitrogen. | [23,24] |
| | *L. acidophilus* (BCRC 10695), *L. delbrueckii* (BCRC 10696), *L. salivarius* (BCRC12574) *Clostridium butyricum* MIYAIRI 588 | The substrate was inoculated with a 3% inoculum and incubated in a chamber at 37 °C for 2–6 days, depending on the assay. Humidity was 40, 45, and 50%. | Soybean oligosaccharides, including raffinose and stachyose, were more efficiently degraded at the initial moisture of 50% compared with other initial moisture contents. Increased levels of the lactic acid, decreased level of pH, and reduced sugar content. | [25] |
| Rapeseed cake | *Aspergillus niger* (CICC 41258) | Solid fermentation at 25, 28, 31, 34, or 37 °C (depending on the test) for 72 h, humidity 43, 50, 56, 62, and 67.5%. | Increased content of crude protein content and the total amino acids (TAA), essential amino acids (EAA), methionine and threonine, no significant difference in lysine. Decrease content of histamine. In vitro TAA and EAA digestibility was improved, the in vitro digestibility of nine amino acids including four essential amino acid (methionine, lysine, arginine, and histamine) also improved. The NDF contents and phityc acid content were reduced but ether extract content increased. | [26] |

**Table 1.** *Cont.*

| Type of Biomass | Microorganisms Used for Fermentation | Process Conditions | Results of Fermentation | References |
|---|---|---|---|---|
| Rapeseed cake | *L. plantarum* LUHS122, *L. casei* LUHS210, *L. farraginis* LUHS206, *P. acidilactici* LUHS29, *L. plantarum* LUHS135, *L. uvarum* LUHS245 | Two-stage fermentation: the first stage was inoculation of the rapeseed cake with a mixture of microorganisms and fermentation for 12 h at 30 °C; in the second stage, 30% of the fermented cake was added to a new batch of rapeseed and fermented for 6 weeks at 30 °C. | Lactic acid bacteria increased, pH decreased, additional essential nutrients were not lost. | [27] |
| Rapeseed middlings (*Brassica napus*), wheat bran (*Triticum eastivum*), two types of brown seaweed (*Saccharina latissima* and *Ascophylum nodosum*) | *Pediococcus acidilactici* (DSM 16243), *Pediococcus pentosaceus* (DSM 12834), *Lactobacillus plantarum* (DSM 12837) | Fermentation for 12 days at 38 °C. | Higher lactic acid content and lower pH. | [28] |
| 40% corn, 40% soybean meal (SBM), and 20% wheat bran | *Bacillus subtilis* ZJU12 | Fermentation at room temperature for 96 h. | Fermented products contained greater concentrations of crude protein, ash, Ca, and total P and more than four times as much Trichloroacetic Acid Soluble Protein (TCA-SP). However, the crude fat decreased. Higher lactic acid content and lower pH. | [29] |
| 12% corn, 20% soybean meal, 48% wheat bran, and 20% soybeans | *Bacillus subtilis* ZJU12, *Pediococcus pentosaceus* ZJUAF-4 | 24 h fermentation of biomass at 37 °C with 40% humidity. | Higher lactic acid content and lower pH. | [30] |
| Wheat and barley | *L. plantarum* DSMZ 8862 and DSMZ 8866, *L. buchneri* NCIMB 40788 | Wheat and barley were milled, inoculated with a 1:1 mixture of *L. plantarum* and *L. buchneri*, and fermented anaerobically for 90 days. Humidity 27%. | Higher lactic acid content and lower pH. | [31] |
| Wheat, barley, and triticale | Natural grain bacteria | The grains of the cereals were mixed with grains of wet wheat stock, whey, and tap water. The mixtures were incubated at 10, 15, or 20 °C. After the 5 days of fermentation, 80% of the contents were replaced with fresh liquids and cereal grains daily for the following 14 days, with 20% left each time as the inoculant for the fresh feed mix. | The cereal grain mix had a more diverse yeast flora, which consisted of *Pichia anomala*, *Rhodotorula glutinis*, *Sporobolomyces ruberrimus*, *Aureobasidium pullulans*, and *Cryptococcus adeliensis*. The LAB *Pediococcus pentosaceus*, *L. plantarum*, *Lactococcus lactis*, and *Lactococcus garvieae* were identified in the cereal grain mix. The LAB population was dominated by *L.plantarum* both before and after storage. The species composition of yeast and LAB populations did not change during grain mix storage. | [32] |

**Table 1.** *Cont.*

| Type of Biomass | Microorganisms Used for Fermentation | Process Conditions | Results of Fermentation | References |
|---|---|---|---|---|
| Wheat, barley, and triticale | Feedtech® F3000 (Delaval International AB, Tumba, Sweden) consisting of a mixture *Enterococcus faecium*, *L.plantarum*, *Lactococcus lactis*, and *Pediococcus pentosaceus* | Feed mixtures after inoculation with lyophilized microorganisms and hydration with tap water were incubated at 20 °C for 5 days. Then, 4/5 of the contents were replaced with fresh compound feed, which was replaced daily for 5 days. The remaining contents served as the inoculum for the next fresh compound feed. | Increased concentrations of all tested organic acids (acetic acid, lactic acid, succinic acid, propionic acid) and ethanol. | [33] |
| Wheat, barley, and soybeans | *Lactobacillus plantarum* DSMZ16627, *Pediococcus acidilactici* NCIMB3005 | The grains were mixed with water. The starter cultures were added, incubated for 48 h at the optimal temperature, and mixed for 30 min with an interval of 30 min between each mixing. | Higher LAB counts. | [34] |

### 2.1. Use of Maize for the Production of Fermented Feed

Corn (*Zea mays*) is a grain grown in North and South America, Asia, Africa, and Europe. About 850 million tons of grain are grown on 162 million hectares of land each year. The USA (37% of world production) and China (21% of world production) are the largest producers of maize. Maize silage is one of the most energetic foods for farm animals, providing high levels of fat and protein. Currently, about two-thirds of the world's maize is used as animal feed [35]. One of the main forms of maize used for animal feed is silage. Maize can be fermented with a high efficiency because it has a high content of soluble sugars. The dry matter content of the whole maize plant silage should be in the range of 32–35%. The parameters determining the nutritional quality of maize silage include the chemical composition and the content of nutrients, such as dry matter, total protein, total sugars, crude fat, crude fiber, NDF, ADF, and starch, as well as minerals such as calcium, phosphorus, potassium, magnesium, sodium, phosphorus, and sulfur [16–18].

Properly produced maize silage should meet the following parameters [18]:

- dry weight: 30–35%;
- starch: minimum 30% in dry matter;
- crude fiber: maximum 20% in dry matter;
- ADF: maximum 25% in dry matter;
- NDF: maximum 45% in dry matter;
- energy content: minimum 6.5 MJ NEL or 0.9 JPM in 1 kg of dry matter.

Corn grain has the highest energy value of any cereal, low fiber content, high concentrations of easily digestible carbohydrates, quite high fat content, and highly digestible nutrients. However, corn kernels have low levels of protein and amino acids such as lysine, tryptophan, and sulfur amino acids (Table 2). The fat in corn kernels is rich in EFAs (Essential Fatty Acids), especially linoleic acid. Unfortunately, due to its high moisture content and content of simple sugars, corn is susceptible to pathogens and mold fungi of the genus *Fusarium* and *Aspergillus*. These fungi produce very dangerous mycotoxins that have a negative impact on the health of animals. Therefore, fermentation has become a

popular method of preserving corn, especially as is about four times less expensive than drying [17,36].

**Table 2.** Nutritional value of corn in pig feed [37].

| Nutrient Ingredient | Nutritive Value (%) |
| --- | --- |
| Protein | 8.80 ± 0.49 |
| Ash | 1.17 ± 0.16 |
| Fat | 3.77 ± 0.48 |
| Total fibre | 12.24 ± 0.93 |
| Insoluble fiber | 11.29 ± 0.85 |
| Soluble fiber | 0.94 ± 0.18 |
| Carbohydrates | 64.77 ± 1.58 |
| Lysine | 2.64 ± 0.18 |
| Methionine | 2.10 ± 0.17 |
| Cysteine | 1.55 ± 0.14 |
| Threonine | 3.23 ± 0.29 |
| Tryptophan | 3.23 ± 0.29 |

*2.2. Use of Yellow and Narrow-Leaved Lupine in the Production of Fermented Feed*

Like maize, lupine is grown all over the world. Poland is the third largest producer of lupine in the world (14.47% of world production), after Australia (47.14% of world production) and Russia (16.51% of world production). The amount of lupine produced in Poland has been increasing steadily since 2018. The most commonly used varieties of lupine bred for the production of animal feed are narrow-leaved husk (*Lupinus angustifolius* L.) and yellow lupine (*Lupinus luteus* L.). Replacing imported soybean feed with legumes is an interesting option for Polish agriculture, since legumes are cultivated locally and have a high crude protein content (40% in yellow lupine seeds). Lupine also has the ability to bind nitrogen, phosphorus, and other elements by releasing citrate into the soil. This is important because the cultivation of lupine can improve the condition of less fertile lands and prevent over exploitation, in accordance with the principles of sustainable agriculture. Lupine tolerates poor and acidic soils well, and is adapted to the temperate climate prevailing in Europe. Lupine has a similar high protein content and amino acid profile to soybeans, but it is less suitable for use in animal feed because of the presence of antinutritional substances, including protease inhibitors, alkaloids, lectins, tannins, and phytates. These substances can be reduced by various thermal treatments, or by the cultivation of sweet lupine varieties or varieties with a low concentration of undesirable substances. Compared to maize silage, lupine silage contains much more protein. Although it is a valuable protein feed, the large amount of protein and small amount of sugars do not favor the fermentation process (Table 3). The use of primers can eliminate this problem. By influencing the rate and course of lactic acid fermentation, primers enable the long-term preservation of the fodder protein without reducing its biological value. To obtain good quality silage, the dry matter content of the green forage should be 28–35% [9,38–41].

*2.3. Other Plant-Based Biomass Used in the Production of Fermented Feed*
2.3.1. Soy

Full fat and high protein (35–40% content [42]) soybeans are a tasty and preferred component in pig feed, both alone and in combination with other vegetable protein sources. Apart from having significantly improved protein digestibility, soybean grains subjected to the extrusion process are also characterized by a better availability of energy. As a result, the fat is better absorbed by the intestinal villi of animals. A fermented soybean meal is

rich in probiotics and functional metabolites, which facilitate digestion, absorption, and the use of soy protein in pigs, as well as inhibiting the growth of pathogens, including *Staphylococcus aureus* and *Escherichia coli* [25].

**Table 3.** Nutritional value of narrow-leaved lupine grains in pig feed.

| Nutrient Ingredient | Concentration in 1 kg of Dry Matter |
|---|---|
| Total protein (%) | 356 |
| Crude fiber (%) | 164 |
| Crude fat (%) | 56 |
| Nitrogen-Free Extracts (%) | 384 |
| Crude ash (%) | 51 |
| Starch (g) | 96 |
| Simple sugars (g) | 54 |

### 2.3.2. Rapeseed

Rapeseed is used in pig feed due to its high protein content (36–38%), which is similar to that in soy. However, rapeseed contains less lysine and more crude fiber than soy. Rapeseed meal is a by-product after oil extraction. Its use in pig feed also reduces problems and costs related to disposal. The extrusion of rapeseed meal makes it highly digestible, thanks to which there is great potential to significantly increase its share in the feed ration for pigs [28,43].

### 2.3.3. Cereals

Since cereals are grown mainly for human consumption, rye and other cereals were for a long time not included as a feed component for pigs. An additional limitation was the ergot content of the old varieties. With the advent of genetically modified rye, its use as a feed ingredient has increased. The high concentration of nonstarch polysaccharides present in rye is undesirable in the diet of young pigs. However, the fermentation of these components contributes to the production of more butyrate and improves intestinal health, providing some prebiotic properties. Wheat is used in the form of decoctions, which are a by-product of ethanol production. This prevents excessive waste and reduces problems and costs related to disposal. Feed with the addition of fermented cereals contributes to reduce the susceptibility of pigs to diarrhea [33,44].

## 3. Biologically Active Substances in Fermented Feed Components

Active substances found in pig feed can be divided into two categories: biologically active substances, the presence of which is desirable because it improves the health of pigs and increases their weight (Table 4); and antinutrients, which have the opposite effects. The antinutrients found in animal feed are mainly derived from feed ingredients. Most occur naturally in plants, protecting them against consumption. They can also come from mold, mycotoxins, or heavy metals (Table 5). Their presence is very undesirable, mainly for economic and health reasons. Antinutrients reduce the palatability of feed, imparting a bitter, tart taste, which may lead to less feed being consumed. Less feed consumption reduces productivity and the quality of the products obtained. The consumption of feed with a high content of harmful antinutritional substances may affect the health of pigs and reduce the safety of meat products. It should be emphasized, however, that it is the dose that makes a substance a poison (as pointed out by Paracelsus), so the mere presence of these substances does not have a negative effect on the feed or pigs. Antinutritional substances are classified into two groups [9]:

- Primary: carbohydrates, proteins, vegetable fats;
- Secondary: protein compounds (phenolic compounds, glucosinolates, glycosides, phytins, or alkaloids).

With the exceptions of oligosaccharides and alkaloids, the varieties of lupine cultivated currently contain similar amounts of antinutritional substances to soy, which facilitates their use in the diet of pigs. Alkaloids are naturally occurring substances produced by plants for defense purposes. Pigs are extremely sensitive to the presence of these poisonous amines in feed. They give the fodder a bitter, unpleasant taste, and may cause disorders of the nervous and digestive systems. With the meat they can enter the human body, where they can also have negative health effects. As a result of the presence of oligosaccharides and alkaloids, feed may not be properly consumed, resulting in economic losses. However, the alkaloids found in the yellow variety of lupine (*L. luteus* L.) are tolerated by young pigs in amounts up to 0.45 g/kg and do not reduce the amount of feed they consume. In comparison, the maximum tolerated level of the alkaloids in white lupine (*L. albus* L.) is more than three times lower. This difference results from the different chemical composition and toxicity of the alkaloids. Narrow-leaved lupine contains very small amounts of alkaloids in its seeds (less than 0.2 g/kg) [9,40].

Carbohydrates with a low molecular weight that are indigestible by hydrolytic enzymes are present mainly in the husks of lupine seeds. One way to reduce their content in lupine feed is to shell the seeds. Shelling the seeds significantly reduces the amount of indigestible fiber and increases the amount of digestible nutrients. This increases the growth efficiency of pigs (mainly by increasing the amount of fecal energy and the digestibility of lysine in the final section of the small intestine) [40,45,46].

**Table 4.** Selected biologically active ingredients present in feed components.

| Biomass | Biologically Active Substance | Effects on Pigs | Biologically Active Substance in Fermented Biomass | Additional Benefit/Effects on Pigs |
|---|---|---|---|---|
| Corn | Polyphenols | Improving antioxidant potential; beneficial effects on lipid metabolism; improving intestinal health [47] | Probiotics | Increasing the natural immunity of pigs; positive effects on offspring; increasing the number of beneficial intestinal bacteria of the genus *Lactobacillus* [19,48,49] |
| | β-glucan, food fiber | Reducing the risk of cardiovascular disease [50] | | |
| | Resistant starch—an insoluble fraction of dietary fiber | Prebiotic; improving intestinal function; reducing symptoms of diarrhea [49] | Lactic acid | Preventing the proliferation of pathogens along the gastrointestinal tract (e.g., *Enterobacteriaceae* such as coliforms and *Salmonella*) [51] |
| | Cathorenoids and flavonoids (anthocyanins) | Reducing the risk of cardiovascular disease in animals [52] | | |
| Lupine | Phenolic antioxidants (caffeic acid and myricetin) | Slowing the oxidation reaction in the body (slowing aging, protects against cancer) [51] | Dietary fiber | Improving the physiology of the gastrointestinal tract and the gut microbiota; may also help maintain intestinal health and prevent postweaning diarrhea [53] |
| Soy | Polyphenols | Antioxidant and fungicidal properties; growth stimulating effects [54] | Free phenolic acids | Antioxidant, antityrosinase, and antiproliferative activities [51] |

**Table 4.** *Cont.*

| Biomass | Biologically Active Substance | Effects on Pigs | Biologically Active Substance in Fermented Biomass | Additional Benefit/Effects on Pigs |
|---|---|---|---|---|
| Soy | Soy isoflavones | Anti-inflammatory, antioxidative properties at cellular levels, engaging several receptors and pathways, including inhibition of nuclear factor kappa-light-chain-enhancer of activated B cells (NF-κB, which plays a key role in regulating the immune response to infection. Disturbances in the regulation of NF-κB are associated with cancer, inflammation and autoimmune diseases, septic shock, viral infections, and inappropriate development of the immune system) activation and inducible-nitric oxide synthase enzymes, thereby ascribing antiviral properties [51] | Flavonoids | Strong antiproliferative activity against cancer cell lines [51] |
| | Soy saponins | Engaging anti-inflammatory pathways [51] | | |
| | Prebiotics (raffinose, stachiosis, inulin, oligofructose) | Stimulating the development of probiotic intestinal flora; reducing the symptoms of hepatic encephalopathy; increasing intestinal peristalsis; lowering the pH and ammonia content in the stool; increasing the amount of short-chain fatty acids [48] | Probiotics | Improving the digestibility of nutrients; improving the composition of the gut microflora of piglets [55] |
| Soybeans, lupins, beans | Lysine (essential amino acid) | Weight gain in pigs [48] | | |
| Rye/rapeseed | Dietary fiber | Improving the physiology of the gastrointestinal tract and the gut microbiota; may also help maintain intestinal health and prevent postweaning diarrhea [56] | | |

**Table 5.** Selected antinutritional components present in feed components.

| Biomass | Antinutritive Substance | Effects on Pigs | References |
|---|---|---|---|
| Legumes | Trypsin inhibitors | Inhibiting the action of trypsin; reducing the digestibility of the protein | [57] |
| Common peas, field beans, and sorghum | Tannins | Protein precipitation; lowering the digestibility of feed | [57] |
| Legumes, cereals | Oligopeptides | Not hydrolyzed in the digestive tract; causing gas and diarrhea | [57] |
| Rapeseed | Glucosinolates | Toxic compounds are formed during decomposition; thyroid hypertrophy; damage to the pancreas and liver | [57,58] |
| Lupins | Alkaloids, lectins, tannins, phytates | Affecting the nervous tissue; damaging the liver | [38,57] |
| | Protease inhibitors, alkaloids, lectins, tannins, or phytates | Reducing digestive capacity and the use of protein by animals (piglets) | [58] |

**Table 5.** *Cont.*

| Biomass | Antinutritive Substance | Effects on Pigs | References |
|---|---|---|---|
| Soy | Antigenic protein (glycinin and β-conglycinin) | Abnormal morphology of the small intestine and diarrhea in rearing piglets | [11] |
| | Trypsin inhibitor, lectin, α-amylase inhibitory factor, and soybean antigens | Reducing the nutritional value, utilization, and digestibility of soybean bios, which can lead to digestive and metabolic diseases | [25] |
| | Phytoestrogens | Negative effects on the reproductive system in sows; negative effects on animal reproduction | [55] |
| Wheat and barley | Phytates | Reducing the digestibility of the feed | [31] |
| Rye | Nonstarch polysaccharides | Not hydrolyzed in the digestive tract; causing gas and diarrhea | [44] |

*Improving the Nutritional Value of Plant Biomass by Lactic Fermentation*

The efficient rearing of pigs is directly influenced by the protein content of the feed ingredients. However, the presence of antinutritional ingredients may contribute to lower protein utilization and digestibility, leading to digestive and metabolic diseases in pigs. The inclusion of highly digestible ingredients in the diet of piglets is extremely important, due to their immature digestive and immune systems. An effective way to eliminate antinutritive compounds is microbial fermentation with the use of probiotic microorganisms. The microorganisms used for fermentation produce organic acids, such as lactic and acetic acids, which significantly lower the pH of the feed (to 3.5–4.5), which inhibits the growth of *Enterobacteriaceae* pathogens in the digestive tracts of animals. Fermentation also contributes to the greater availability of phosphorus in grain diets. When fermented with the help of lactic bacteria, mainly of the *Lactobacillus* genus, pig food components are characterized by a lower content of antinutritional factors, such as dietary fiber and phytates. The lactic acid metabolized by lactic bacterial also has antifungal properties [25].

Another positive aspect of the use of fermented food is the presence of the metabolites of microorganisms and lactic acid bacteria, which, when delivered to the intestines of animals, improve the condition of the intestinal microbiome. This in turn contributes to a more efficient use of energy. It has been shown that soybean meal fermented with probiotics, including species of the genus *Aspergillus*, *Lactobacillus* (including *L. plantarum*, *L. acidophilus*, *L. delbrueckii*, and *L. salivarius*), and *Clostridium butyricum*, do not cause diarrhea in pigs or piglets weaned from sows [25]. Soybean meal fermented with probiotics is also more digestible by animals and improves their growth. Some *Lactobacillus* species, which produce lactic acid through anaerobic fermentation, remove the trypsin inhibitor that reduces the amount of protein that the digestive system can absorb from food. This contributes to increased protein hydrolysis and the release of free amino acids. The probiotic spore-forming *C. butyricum* strain contributes to improving the growth efficiency of pigs and their immunity [25,31,33].

The interest of the livestock industry in fermentation is due primarily to the fact that fermentation can provide health and environmental benefits at a low cost. A study by Fan et al. [16] showed that feeding pigs with fermented corn–soybean meal significantly increased their daily weight gain and food intake. The fermented feed also increased the transcription of insulin-like growth factor 1 (IGF1) in the liver and serum of pigs, which stimulates growth. The process of fermenting feed degrades the antinutritional components present in soybean meal substantially, increasing the digestibility of amino acids and phosphorus. This contributes to reduce emissions to the environment. Feeding pregnant sows with fermented corn and soybean meal has been shown to increase the weight of their offspring. Research on the mechanism of growth stimulation in pigs fed with fermented fodder is being conducted in parallel with research on the improvement of

the fermentation process, to reduce the antinutrients in maize to levels similar to those in soybean feed [16].

Rho et al. [59] showed that fermented maize improves the growth of young pigs in the first 3 weeks after starting silage feeding. However, the feed fermentation process is complicated and should be constantly monitored and optimized [59]. Lin et al. [19] investigated corn cobs fermented with prebiotics as feed for fattening pigs. The results showed an improvement in the appearance and texture of the prepared feed, as well as in the quality of the nutrients. The amount of consumed feed increased, along with the rate of nutrient digestion and weight gain. The amount of beneficial intestinal bacteria of the genus *Lactobacillus* also increased. The decrease in the amount of pathogenic bacteria, including *Escherichia coli*, improved the composition of the intestinal microbiota of fattening pigs and increased their resistance to pathogens [19]. The effect of fermented soybean corn meal on the health and immunity of pigs was examined by Lu et al. [53]. Their research showed an increase in the amount of IgG and IgM immunoglobulins in groups of fattening pigs fed with fermented fodder. Increased IgM concentration improved immune status, and IgG is an indicator of the immune status. The levels of both parameters were significantly higher in the research groups, which suggests that the consumption of fermented feed with the addition of maize or with maize alone improves the level of pig immunity. This contributes to less frequent infections in fattening pigs and reduces economic losses [53].

Hao et al. [30] demonstrated that the addition of fermented feed to pig diets had positive effects on food intake, nutrient utilization, and intestinal health. The probiotics added to the feed also had a positive effect on the intestinal microbiota. The presence of the probiotic *Lactobacillus* planetarium strain used by Yang et al. [60] not only improved weight gain, nutrient digestibility, and piglet fecal microbiota, but also reduced harmful gas emissions in weaned piglets [30,60].

Cebulska et al. [61] investigated the effects of replacing soy protein in pig diets with pea protein and yellow lupine. Compared with the animals in the control sample, the meat from legume-fed pigs had a more favorable proportion of exogenous amino acids (except methionine), and higher amounts of micronutrients such as zinc and iron. Phosphorus and potassium were noted as more common macronutrients. Insignificant differences were noted in the amounts of amino acids in the experimental groups, indicating that replacing soybean in feed with legumes does not adversely affect the appearance, physicochemical quality, amino acid profile, or fat profile of pig meat. Zaworska-Zakrzewska et al. [46] showed that the 24 h fermentation of lupine with bacteria and yeast increases the content of protein, crude fiber, and ash. The amino acid profile was similar to raw lupine. However, the content of oligosaccharides and phytates decreased, the pH also decreased, and the level of alkaloids remained the same. There were no significant differences between the of meat obtained from pigs fed with fodder with the addition of raw lupine, fermented lupine, or fodder with soy. Similar results were reported by Cebulska et al. [61], showing that lupine can replace soy in food intended for pigs.

## 4. Conclusions

The introduction of fermented plant-based biomass to enrich fodder is a beneficial solution from the point of view of the circular economy and the improvement of animal welfare. Corn and lupine are attracting attention as a source of biomass that will be fermented and then used as pig feed. The reason for this attention is their ability to ferment effectively, the possibility of using maize and lupine from waste, and the numerous biologically active substances contained in them. The fermentation of corn and lupine allows for a higher amount of crude protein in pig meat. The appearance and taste of the meat improve, and the amount of water in the meat decreases. The lower content of antinutritional substances and the improved digestibility of the silage also contribute to improve the composition of the intestinal microflora of pigs. Importantly, it is unnecessary to use antibiotics as growth promoters with fermented feed, reducing the spread of antibiotic resistance.

**Author Contributions:** B.P., I.M., J.B. and M.M.-F. contributed to the conception and preparation of the manuscript and critical revision. J.B., I.M. and B.P. participated in the review and final editing of the manuscript. All authors have read and agreed to the published version of the manuscript.

**Funding:** This research received no external funding.

**Institutional Review Board Statement:** Not applicable.

**Informed Consent Statement:** Not applicable.

**Data Availability Statement:** Not applicable.

**Conflicts of Interest:** The authors declare no conflict of interest.

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
