# Peer review of "The Use of Fermented Plant Biomass in Pigs Feeding"

_sustainability, doi:10.3390/su142114595_

Round 1

Reviewer 1 Report

See the attached.

Reviewer 2 Report

The authors shall make some comments in the conclusion for the future research.

Reviewer 3 Report

 The authors did not provide the analytical calculations based on which they made the calculations. The article contains more generalized information not confirmed by own calculations. In my opinion, this is a simple review and synthesis of two methods, not confirmed by an experimental method.

«According to UN predictions, the human population is expected to reach almost 10 50 billion by 2050. Population growth is creating increased demand for meat products. At 51 the same time, consumer standards are rising. Meat obtained by breeding animals, includ- 52 ing pigs, must meet higher expectations, not only in terms of quality but also of appear- 53 ance. This demand creates a shortage of animal feed, and contributes to the growing in- 54 terest in the modification of plant waste into wholesome animal food.» - known statement, no reference source indicated.

The authors are requested to describe the algorithm of the performed calculations, to provide experimental studies in graphical or analytical form. And make these changes in the research methods section.

Author Response

According to UN predictions, the human population is expected to reach almost 10 billion by 2050. Population growth is creating increased demand for meat products. At the same time, consumer standards are rising. Meat obtained by breeding animals, including pigs, must meet higher expectations, not only in terms of quality but also of appearance. This demand creates a shortage of animal feed, and contributes to the growing interest in the modification of plant waste into wholesome animal food [2]. - reference source indicated.

Round 2

Reviewer 1 Report

The authors should have submitted a revised version of the manuscript without track changes for ease of review. Otherwise, the highlighted comments have been addressed.